# The Heterogeneous Discrete Generalized Nash Model for Flood Routing

Baowei Yan[1,*], Huining Jiang [1], Zhengkun Li[1], Jun Zhang[2], Wenfa Yang[2]

[1]School of Hydropower and Information Engineering, Huazhong University of Science and Technology,

Wuhan 430074, China

[2]Bureau of Hydrology, Changjiang Water Resources Commission, Wuhan 430010, China

*Correspondence to: Baowei Yan (bwyan@hust.edu.cn)*

**Abstract.** The topographic heterogeneity of the rivers has great effects on the river flood routing. The

discrete generalized Nash model (DGNM), developed on the basis of the Nash's instantaneous unit

hydrograph (IUH), is a lumped model that can't reflect the spatial heterogeneity of the river topography.

The heterogeneous DGNM (HDGNM) with a consideration of such heterogeneity has been developed

by the conceptual interpretation of the DGNM. Two compositions of the downstream outflow

generated by the recession of the old water stored in the river channel and the discharge of the new

water from upstream inflow were deduced respectively with the help of the heterogeneous IUH and

the corresponding heterogeneous S curve. The HDGNM is finally expressed as a linear combination

of the inflows and outflows, whose weight coefficients are calculated by the heterogeneous S curve.

The HDGNM expands the application scope, and becomes more applicable, especially in river reaches

where the river slopes and cross-sections change greatly. The middle Hanjiang River was selected as

a case study to test the model performance. It is suggested that the HDGNM performs better than the

DGNM, with higher model efficiency and smaller relative error in the simulated flood hydrographs.





## 1 Introduction

In rainfall-runoff modelling, the instantaneous unit hydrograph (IUH) proposed by Nash (Nash, 1957) is one of the most widely used methods for overland flow routing. Under the assumption that

watershed was represented as a cascade of equal linear reservoirs, IUH was obtained in a form of gamma distribution with two parameters – n, the number of linear reservoirs, and K, reservoir storage coefficient. The linear cascade concept has greatly promoted the development of the flow routing theory. Many linear cascade – based models have been developed since then. However, as a lumped model, IUH cannot reflect the spatial heterogeneity of rainfall and landforms. Great efforts have been

made to make IUH be semi-distributed or distributed mainly by two approaches. The first approach has been mostly performed by replacing the equal reservoirs in the IUH with unequal ones. Dooge (1959) conceptualized the watershed as a combination of unequal linear reservoirs and linear channels, and developed a general theory for unit hydrograph. Singh (1964) derived the IUH using a nonlinear model considering the overland and channel flow components separately, in which two unequal linear

reservoirs with different storage coefficients were used. To solve the flow routing in urban areas, Diskin et al. (1978) proposed an urban parallel cascade IUH model by representing the basin system as the combination of two parallel branches having a series of equal linear reservoirs. Bhunya et al. (2005) developed a hybrid model by splitting the single linear reservoir into two serially connected reservoirs of unequal storage coefficients (one hybrid unit), and obtained the analytical expression of

the model for two hybrid units in series. Later, to consider the translation time, Singh et al. (2007) extended this hybrid model by inserting a linear channel between each hybrid unit. Bhunya et al. (2008) formulated a rainfall–runoff model incorporating a variable storage coefficient instead in the two-

reservoir Nash cascade model. Li et al. (2008) derived the IUH with different $K$ values for each

reservoir using the Laplace transform and developed a general rule for the equation of the IUH of any

order. The other approach is to divide the watershed into a number of subwatersheds to consider the

nonuniformly distributed rainfall. For example, based on the structure of a stream network, Wang and

Chen (1996) divided the watershed into a number of subwatersheds and obtained the outflow

hydrograph of each subwatershed based on the concept of linear cascade reservoirs. This linear,

spatially distributed model can be capable of predicting runoff from non-uniformly distributed rainfall

and geographical conditions over an entire watershed. Similarly, Wan et al. (2016) divided the

watershed into subareas by isochrones, and established an independent linear reservoir-channel

cascade model in each subarea. Finally, the generalized concentration curve that can be applied to large

heterogeneous watersheds was derived. All of these modifications of IUH have made a certain

improvement in the rainfall-runoff modeling.

55        As a general method of flow routing, Nash's IUH is also applicable to river flow routing, which

has been done independently by Kalinin and Milyukov (1958), also known as Characteristic Reach

method. Nash's IUH can also be obtained by solving the $n$th order differential equation of a linear

system with the zero initial conditions (Chow, 1988). Zero initial conditions represent that the linear

reservoirs in the Nash cascade model are empty at initial time, or equivalently the initial river storages

are empty when IUH is applied in river flow routing, which does not match the fact. To consider the

influence of the initial state, Szollosi-Nagy (1982) formulated a state-space description of the Nash

cascade model in a matrix form, whereby the initial storage of the river system should be estimated

separately via observability analysis (Szollosi-Nagy, 1987). Szilagyi (2003) then extended this model





to a sample-data system framework and made some modifications to make it more applicable (Szilagyi,

2006; Szilagyi and Laurinyecz, 2014). Recently, Yan et al. (2015) exactly solved the *n*th order

differential equation of the Nash cascade model with the same non-zero initial condition, and obtained

the generalized Nash model (GNM) with a simpler expression, in which the initial state was directly

included and should not be estimated separately anymore. To make the GNM be applied easily to the

sample-data system, Yan et al. (2019) further discretized its analytical expression by introducing a

variable $S_n$-curve, and obtained the discrete generalized Nash model (DGNM). The DGNM expresses

the outflow as a linear combination of the old water stored in the river reach and new water from the

upstream inflow. The DGNM is based on the lumped IUH that it cannot reflect the spatial

heterogeneity of the river topography too. So the DGNM is less applicable for those the topography

changes large along the river. In this case, the unequal reservoir concept should be used instead.

However, under the non-zero initial conditions, the solving of high order differential equation of the

Nash cascade model with unequal storage parameters will become very difficult, which makes the

generalization of the Nash model impossible by directly solving the differential equation. A new way

is proposed in this paper to obtain the heterogeneous DGNM (HDGNM) through the conceptual

interpretation and mathematical derivation of the DGNM.

## 2 Conceptual interpretation of the DGNM

According to Yan et al. (2019), the calculation formula of the DGNM is

$$O_{t+1} = \sum_{j=0}^{n-1}\sum_{i=0}^{j} \frac{(-1)^i}{j!} C_j^i \left(1 - S_{n-j}\right) O_{t-i} + S_n I_t + \left(1 - \frac{K}{\Delta t}\sum_{i=1}^{n} S_i\right)\Delta I_{t+1} \tag{1}$$





Where $C_j^i$ is the combination formula; $O_{t\pm i}$ represents the downstream outflow at time $t \pm i\Delta t$; $\Delta t$ is the time interval; $I_t$ represents the upstream inflow at time t; $\Delta I_{t+1}$ represents the inflow increment

during the time interval $[t, t + \Delta t]$; n and K are model parameters; $S_i$ can be computed by

$$S_i = 1 - e^{-\frac{\Delta t}{K}} \sum_{j=0}^{i-1} \frac{1}{j!} \left( \frac{\Delta t}{K} \right)^j \qquad (2)$$

According to the definition of the S curve, $S_i$ represents the outflow generated by the unit continuous inflow after the routing of a series of i reservoirs at the end of the time. The calculation formula of the DGNM shows that the downstream outflow is composed of three items. The first item

is the recession flow of the current water storage capacity in the channel, which is the superposition of the flow generated by the current storage of each reservoir routed by the subsequent reservoirs. The second term is the recession flow generated by the current inflow $I_t$ routed by river channel, or equivalently by a series of n cascade linear reservoirs. According to the definition of $S_i$ as well as the storage-discharge relation of the linear reservoir, $KS_i$ represents the water stored in each reservoir for

a unit continuous inflow, and $\sum_{i=1}^{n} K S_i / \Delta t$ represents the ratio of water stored in the channel during the period $\Delta t$. Then, $1 - \sum_{i=1}^{n} K S_i / \Delta t$ represents the ratio of water discharges from the channel. So the third term is the outflow generated by the inflow increment during the time interval $[t, t + \Delta t]$ after the channel routing. In summary, the downstream outflow is generated by the old water stored in the river channel and the new water from upstream inflow. Part of the new water flows out of the

downstream section and becomes one part of the outflow, the other part remains in the river channel to supplement the old water. The old water recedes and becomes the other component of the outflow. In such circulation, the outflow process of the downstream section is formed. Through the conceptual interpretation of the DGNM, the downstream outflow is physically generated by the old water stored





in the river reach and new water from the upstream inflow, and formally expressed as a linear

combination of the inflows and outflows, whose weight coefficients are calculated by the *S* curve that

gives another way to deduce the HDGNM by introducing the heterogeneous S curve.

**3 Heterogeneous S curve**

The routing storage capacity of the basin is affected by geographical features and has spatial

heterogeneity. The storage routing effect of the basin is equated to a cascade reservoir routing in Nash's

IUH, and each reservoir has the same storage coefficient. This generalization has certain rationality

for basins with homogeneous topography variation. But for the basins with large topographic changes,

the spatial difference in storage routing effect will be more significantly affected by topography and

geomorphology. To consider such spatial heterogeneity in storage routing effect, Dooge (1959)

conceptualized the watershed as a combination of unequal linear reservoirs and linear channels, and

developed a general theory for unit hydrograph. Li et al. (2008) further deduced the IUH with different

storage parameters, here we call it heterogeneous IUH (HIUH) to distinguish with Nash' IUH:

$$u_n(t) = \sum_{j=1}^{n} \frac{K_j^{n-2}}{\prod_{i=1, i \neq j}^{n} (K_j - K_i)} e^{-\frac{t}{K_j}} \tag{3}$$

where $K_i (i = 1, \cdots, n)$ is the storage parameter of the i-th reservoir (the numbers here are sorted in

the forward direction, that is, the most upstream reservoir is the number 1, and the most downstream

reservoir is the number n). Correspondingly, the heterogeneous S curve formed by HIUH is (Li et al.,

2008)

$$S_n(t) = \int_0^t u_n(t) dt = \sum_{j=1}^{n} \frac{K_j^{n-1}}{\prod_{i=1, i \neq j}^{n} (K_j - K_i)} \left( 1 - e^{-\frac{t}{K_j}} \right) \tag{4}$$





where $S_n(t)$ represents the outflow of the *n*th reservoir yielded by a continuous unit upstream inflow.

If further define the storage curve, we obtain

$$R_n(t) = \int_t^{+\infty} u_n(t)\,dt = \sum_{j=1}^{n} \frac{K_j^{n-1} e^{-\frac{t}{K_j}}}{\prod_{i=1, i \neq j}^{n} \left( K_j - K_i \right)} \tag{5}$$

where $R_n = 1 - S_n(t)$, represents the detention storage of the *n*th reservoir yielded by a continuous unit upstream inflow.

The HIUH is a more accurate generalization of the watershed storage routing, and is a theoretical expansion of the Nash's IUH. With consideration of the spatial heterogeneity in the storage routing, HIUH is especially applicable to the basin with large topographic changes. The DGNM is developed on the basis of the Nash's IUH, which leads to its theoretical limitations when applied to the river reach with large changes of cross-sections and slopes. The introduction of HIUH can reflect the difference of flood routing in each sub-river, thus can improve the flood simulation precision in the whole river reach theoretically.

**4 Derivation of the heterogeneous DGNM**

The conceptual interpretation of the DGNM shows that the downstream outflow is generated by the old water stored in the channel and the new water from upstream inflow, denoted by $O^{old}$ and $O^{new}$ respectively, we have

$$O_{t+1}^{old} = \sum_{j=0}^{n-1} \sum_{i=0}^{j} \frac{(-1)^i}{j!} C_j^i \left( 1 - S_{n-j} \right) O_{t-i} \tag{6}$$

$$O_{t+1}^{new} = S_n I_t + \left( 1 - \frac{K}{\Delta t} \sum_{i=1}^{n} S_i \right) \Delta I_{t+1} \tag{7}$$



For the linear reservoir system with unequal storage parameters, $S_i$ represents the outflow of the $i$th reservoir yielded by a continuous unit upstream inflow, then $K_i S_i$ represents the water stored in each reservoir for a unit continuous inflow, and $\sum_{i=1}^{n} K_i S_i / \Delta t$ represents the ratio of water stored in the channel during the period $\Delta t$, based on the former conceptual interpretation of the $O^{new}$. Hence,

for the linear reservoir system with unequal storage parameters, the outflow generated by "new water" can be obtained by replacing the storage parameter K and S curve in Eq. (7) with variable $K_i$ and heterogeneous S curve, respectively, i. e.

$$O_{t+1}^{new} = S_n I_t + \left(1 - \frac{1}{\Delta t}\sum_{i=1}^{n} K_i S_i\right)\Delta I_{t+1} \qquad (8)$$

Therefore, on the basis of the conceptual interpretation of the DGNM, the outflow $O^{new}$ formed

by the new water can be deduced directly. But it seems impossible to obtain the outflow $O^{old}$ by directly using the heterogeneous S curve instead of in Eq. (7) due to the coefficient of $S_{n-j}$ is also varying with it. For the sake of simplicity, we assume that the most downstream reservoir is numbered 1, and the most upstream reservoir is numbered n, that is to say, the n reservoirs are reversely numbered. Then the storage routing equation of the j-th reservoir can be obtained from the water balance equation:


$$K_j \frac{dO_j(t)}{dt} = O_{j+1}(t) - O_j(t) \qquad (9)$$

It can be known from Eq.(9) that the outflow of each reservoir at the current time is as follows:

$$O_1(t) = O(t)$$

$$O_2(t) = O(t) + K_1 O'(t)$$

$$O_3(t) = O(t) + (K_1 + K_2) O'(t) + K_1 K_2 O''(t)$$


$$\cdots\cdots\cdots$$


$$O_j(t) = O(t) + \sum_{p=1}^{j-1} \prod_{r_p > \cdots > r_1 = 1}^{j-1} \left( K_{r_1} \cdots K_{r_p} \right) O^{(p)}(t) \qquad (10)$$

Based on the physical interpretation of the GNM (Yan et al., 2015), the recession flow of the current water storage in river channel is the superposition of the recession flow generated by the current water storage in each reservoir. According to the conception of linear reservoir, the current water

storage of the j-th reservoir is $K_j O_j(t)$, which can be treated as an instantaneous inflow into each reservoir, then the outflow at the end of the period generated by that is $K_j O_j(t) u_j(\Delta t)$. Based on the principle of superposition, the outflow at the end of the period formed by the current water storage of all reservoirs is

$$\mathrm{O}_{t+1}^{old} = \sum_{j=1}^{n} K_j O_j(t) u_j(\Delta t)$$


$$= \sum_{j=1}^{n} K_j u_j(\Delta t) \left[ O(t) + \sum_{p=1}^{j-1} \prod_{r_p > \cdots > r_1 = 1}^{j-1} \left( K_{r_1} \cdots K_{r_p} \right) O^{(p)}(t) \right] \qquad (11)$$

The formula shows that the recession process can finally be expressed as a linear combination of $0 \sim (n-1)$ derivatives of the current time $O(t)$, which is

$$\mathrm{O}_{t+1}^{old} = \sum_{p=0}^{n-1} A_p O_t^{(p)} \qquad (12)$$

where $A_p (p = 0, \cdots, n-1)$ is the coefficient of p-th order derivative of $O(t)$, then we have

(detailed derivation is provided in Appendix)

$$A_p = \begin{cases} 1 - S_n, & p = 0 \\ \displaystyle\sum_{r_p > \cdots > r_1 = 1}^{n-1} \left( K_{r_1} \cdots K_{r_p} \right) \left( S_{r_p} - S_n \right), & p > 0 \end{cases} \qquad (13)$$

$A_p$ is derived in the case of reverse numbering, and its calculation formula shows that $K_i$ has symmetry. Therefore, in the case of forward numbering, or equivalently, let $K_1, \cdots, K_n$ replace





$K_n, \cdots, K_1$ respectively, the formula for calculating $A_p$ is still the same as Eq.(15), thus ensuring that

$O^{old}$ and $O^{new}$ are calculated under the same numbering system. To further discretize the term $O^{old}$

in Eq. (12), the derivatives of $O(t)$ are approximated by the following backward finite difference

method.

$$O_t^{(p)} = \frac{1}{\Delta t^p} \sum_{i=0}^{p} (-1)^i C_p^i O_{t-i} \tag{14}$$

Substituting equation (13) and (14) into equation (12), we obtain


$$O_{t+1}^{old} = \sum_{p=0}^{n-1} \sum_{i=0}^{p} \frac{(-1)^i}{\Delta t^p} C_p^i A_p O_{t-i} \tag{15}$$

According to the conceptual interpretation of the DGNM, the downstream outflow is jointly

produced by the old water stored in the river channel and the new water from upstream inflow, then

we have

$$O_{t+1} = \sum_{j=0}^{n-1} \sum_{i=0}^{j} \frac{(-1)^i}{\Delta t^j} C_j^i A_j O_{t-i} + S_n I_t + \left(1 - \frac{1}{\Delta t} \sum_{i=1}^{n} K_i S_i\right) \Delta I_{t+1} \tag{16}$$

Eq. (16) is the calculation formula of HDGNM, and is also the discrete solution of linear cascade

model with unequal reservoirs under non-zero initial conditions. Like the HIUH, with a consideration

of the spatial heterogeneity in the storage routing, HDGNM should be more applicable to the river

reach with large changes of cross-sections and slopes.

**5 Case study**

To test the applicability of the HDGNM, the river reach between gauging stations Huangjiagang

and Xiangyang in the Hanjiang River of China is selected as a case study. Hanjiang River is one of the

most important tributaries of the Yangtze River of China. The Danjiangkou reservoir, located on the



upper Hanjiang River, is the water source of the middle route of China's south-to-north water transfer

project. Since the opening of the first phase of the middle route on Dec. 12, 2014, the Danjiangkou

reservoir has supplied a total of 25.5 billion cubic meters of water to drier north areas in China, thereby

benefiting 58 million people in Beijing, Tianjin, Hebei province, and Henan province. The studied

river reach with a length of 105 kms is located in the middle Hanjiang River, where the Huangjiagang

hydrological station is located at 6 kms downstream of the Danjiangkou dam site and serves as the

outflow control station of the Danjiangkou reservoir. The interval basin area between Huangjiagang

and Xiangyang is 8044 $km^2$. The sketch map of the middle Hanjiang River and the studied river reach

are shown in Fig. 1. The studied river reach is located in the hilly and plain areas, where hills, terraces,

artificial narrows and wide valleys distribute alternatively, and showing obvious lotus root node shape

on the plane. The main channels in wide sections have large swings and many beaches, but become

single in narrow sections, which makes a large change of the shape in the sections along the river reach,

as shown in Fig.1. The mean slopes of sub-reaches Huangjiagang - Guanghua, Guanghua - Taipingdian,

Taipingdian - Niushou and Niushou - Xiangyang are 0.000176, 0.000276, 0.000221 and 0.000214

(Gong, 1982), respectively. It is indicated that the channel slope of the studied river reach changes

largely, especially from sub-reach 1 to sub-reach 2. In short, the topography of the studied river reach

varies greatly. For such significant spatial heterogeneity, the proposed HDGNM should be more

applicable as interpreted above.

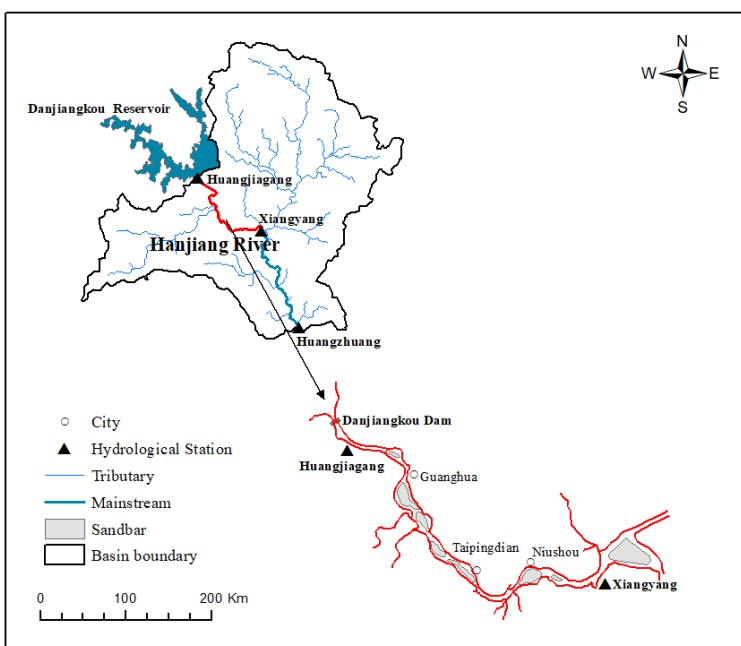

**Fig.1** The sketch map of the middle Hanjiang River and the studied river reach

According to the flood data of the Huangjiagang and Xiangyang hydrological stations from the
year 1974 to 2011, 10 floods with a low proportion of the lateral inflows (time interval $\Delta t = 3h$) were
selected, of which 8 floods were used for model calibration, and the other 2 floods were used for
validation. In order to demonstrate the simulation effect and test the forecast capability of the HDGNM,
the DGNM was selected for comparison. For application, the model parameters in these two models
should be calibrated by using the observed flood data. The SCE-UA algorithm, with the advantages of
robust and reliable performance, global search capability, has become a commonly used optimization
method for hydrological model parameters (Duan et al., 1994), and hence was used to obtain the
optimized parameters of these two models. Take the root mean square error of the observed 8 floods
from the year 1974 to 2005 as the objective function, and run the SCE-UA algorithm to mimimize this
objection function, we obtain the optimized parameters of $n = 3$, $K = 3.51h$ for the DGNM and





n = 3, $K_1 = 1.58h$, $K_2 = 8.80h$, $K_3 = 1.59h$ for the HDGNM. The parameter K reflects the

difference in the storage capacity of linear reservoirs. The three linear reservoirs of the DGNM are

equivalent, in essence, it is a homogenization of the topographical differences of the sub-reaches.

While two of the three linear reservoirs of the HDGNM are approximately the same, and the other is

quite different from the two. Therefore, it can more objectively reflect the influence of topographical

differences on the storage routing of the river channel. Theoretically, it can improve the accuracy of

the flood routing in river channel. The performances of these two models were assessed by the

following two commonly used statistics (Wu et al. 2012):

(1) Relative error (*RE*) of peak discharge

$$RE = \frac{\left|O_{p,est} - O_{p.obs}\right|}{O_{p.obs}} \times 100\% \tag{17}$$

in which $O_{p,est}$ and $O_{p,obs}$ are the estimated and observed peak discharge, respectively.

(2) Nash–Sutcliffe efficiency coefficient (*$E_{NS}$*).

$$E_{NS} = 1 - \frac{\sum\limits_{t=1}^{T}(O_{t,est} - O_{t,obs})^2}{\sum\limits_{t=1}^{T}(O_{t,obs} - \bar{O}_t)^2} \tag{18}$$

in which $O_{t,est}$ and $O_{t,obs}$ are estimated and observed discharge at time *t*, respectively. $\bar{O}_t$ represents

the mean of observed discharge.

**Table 1** The accuracy evaluation results of DGNM and HDGNM

| Period | Data | DGNM | | HDGNM | |
|---|---|---|---|---|---|
| | | *RE*(%) | *$E_{NS}$* | *RE*(%) | *$E_{NS}$* |
| | October 1974 | 0.91 | 0.9930 | 4.27 | 0.9861 |
| | September 1975 | 3.69 | 0.9918 | 2.87 | 0.9950 |
| Calibration | July 1980 | 1.62 | 0.9043 | 1.08 | 0.9284 |
| | August 1981 | 5.26 | 0.9699 | 3.47 | 0.9747 |
| | September 1981 | 7.82 | 0.9655 | 4.45 | 0.9806 |





|  |  |  |  |  |  |
| --- | --- | --- | --- | --- | --- |
|  | September 1984 | 0.99 | 0.9774 | 1.78 | 0.9860 |
|  | September 2003 | 2.54 | 0.9655 | 0.77 | 0.9715 |
|  | October 2005 | 8.72 | 0.9845 | 0.63 | 0.9963 |
|  | Average | 3.94 | 0.9690 | 2.42 | 0.9773 |
| Validation | July 2007 | 0.62 | 0.9707 | 0.66 | 0.9775 |
|  | September 2011 | 2.48 | 0.9889 | 0.32 | 0.9928 |
|  | Average | 1.55 | 0.9798 | 0.49 | 0.9852 |

To further test the validity of the parameter estimations from calibration, the verification experiment was also conducted. The other 2 observed floods in the year 2007 and 2011 were adopted to verify the calibration results. The accuracy evaluation results of these two models in calibration and validation periods were both shown in Table 1. In calibration period, the average values of $RE$ and $E_{NS}$ were 3.94% and 0.9690 for DGNM, respectively. Compared with the DGNM, the HDGNM has made

some improvements in the simulation. The average value of $RE$ has reduced to 2.42% and that of $E_{NS}$ has increased to 0.9773. The similar improvements can be found in the validation period with values from 1.55% to 0.49% for $RE$ , and from 0.9798 to 0.9852 for $E_{NS}$, respectively. Comparison of the observed and simulated hydrographs for the selected 10 floods was shown in Fig. 2. The HDGNM with a consideration of the topographical heterogeneity of the river reach, makes the simulated

hydrographs much closer to the measured flood hydrographs, especially near the flood peaks. Except for the October 1974 flood event, the simulations of other floods have different improvements. The simulation results show that, compared with the DGNM, the spatial difference of the storage parameter is considered, thus the HDGNM is more applicable, especially in the river reaches where the river slopes and cross - sections change greatly.




**Fig. 2** Comparison of the observed and simulated hydrographs for the selected 10 floods

## 6 Conclusions

The heterogeneous DGNM for flood routing was deduced indirectly by conceptual interpretation of the DGNM. It is suggested that the downstream outflow is generated by the recession of the old water stored in river channel and the discharge of the new water from upstream inflow, and can

formally expressed as a linear combination of the inflows and outflows, whose weight coefficients are calculated by the $S$ curve. When such old water and new water are routed by a series of unequal reservoirs, the DGNM becomes to HDGNM. Hence, these two compositions of the outflow were deduced respectively. The discharging part produced by the new water can be easily deduced with the help of the heterogeneous S curve. The recession part produced by the old water is obtained by the

superposition of the recession process for each linear reservoir, which can be calculated by the impulse response of the current stored water with the help of HIUH. At last, the HDGNM is expressed as a linear combination of the inflows and outflows, whose weight coefficients are calculated by the heterogeneous $S$ curve.

The proposed HDGNM was applied to a reach of the middle Hanjiang River with large changes

of river slopes and cross-sections. A different linear reservoir with a much larger storage coefficient was detected from the conceptualized reservoirs in the studied river reach. Considered of the topographical heterogeneity of the river reach, the HDGNM performs better than the DGNM, with higher model efficiency and smaller relative error in the simulated flood hydrographs. The HDGNM enriches the existing generalized Nash flow routing theory and becomes more applicable, especially

in river reaches where the river slopes and cross-sections change greatly. The river storage is conceptually as a series of unequal linear reservoirs, thus the HDGNM may have the potential for



semi-distributed modelling, e.g. river flow routing with multiple tributaries inflows, which will be

further studied in the future.

**Acknowledgments**

This study is financially supported by the National Key R&D Program of China

(2016YFC0402708), the project of Power Construction Corporation of China (DJ-ZDZX-2016-02),

and the Fundamental Research Funds for the Central Universities (HUST: 2017KFYXJJ195).

**Data Availability**

The data that support the findings of this study are available from the corresponding author upon

reasonable request.

**Author contributions**

Baowei Yan contributed to the development of the model and revised the manuscript. Huining

Jiang wrote the first draft of the manuscript. Zhengkun Li collected and preprocessed the data. Jun

Zhang and Wenfa Yang provided recommendations for the data analysis, participated in discussions

about the results.

**Competing interests**

The authors declare that they have no competing interests.

**Appendix: Derivation of the Coefficient A_P**

In the derivation of the coefficient $A_P$, the following identity holds for any integer $m \in (1, n]$ and

any time t

$$K_m u_m(t) = R_m(t) - R_{m-1}(t) \qquad (A.1)$$

The proof is as follows





$$R_m(t) - R_{m-1}(t) = \sum_{j=1}^{m} \frac{K_j^{m-1} e^{-\frac{t}{K_j}}}{\prod_{i=1,i\neq j}^{m} (K_j - K_i)} - \sum_{j=1}^{m-1} \frac{K_j^{m-2} e^{-\frac{t}{K_j}}}{\prod_{i=1,i\neq j}^{m-1} (K_j - K_i)}$$

$$= \frac{K_1^{m-1} e^{-\frac{t}{K_1}}}{(K_1 - K_2)\cdots(K_1 - K_m)} + \cdots + \frac{K_{m-1}^{m-1} e^{-\frac{t}{K_{m-1}}}}{(K_{m-1} - K_1)\cdots(K_{m-1} - K_m)} + \frac{K_m^{m-1} e^{-\frac{t}{K_m}}}{(K_m - K_1)\cdots(K_m - K_{m-1})}$$

$$- \frac{K_1^{m-2} e^{-\frac{t}{K_1}}}{(K_1 - K_2)\cdots(K_1 - K_{m-1})} - \cdots - \frac{K_{m-1}^{m-2} e^{-\frac{t}{K_{m-1}}}}{(K_{m-1} - K_1)\cdots(K_{m-1} - K_{m-2})}$$

$$= \frac{K_m K_1^{m-2} e^{-\frac{t}{K_1}}}{(K_1 - K_2)\cdots(K_1 - K_m)} + \cdots + \frac{K_m K_{m-1}^{m-2} e^{-\frac{t}{K_{m-1}}}}{(K_{m-1} - K_1)\cdots(K_{m-1} - K_m)} + \frac{K_m K_m^{m-2} e^{-\frac{t}{K_m}}}{(K_m - K_2)\cdots(K_m - K_{m-1})}$$

$$= K_m \sum_{j=1}^{m} \frac{K_j^{m-2} e^{-\frac{t}{K_j}}}{\prod_{i=1,i\neq j}^{m} (K_j - K_i)}$$

$$= K_m u_m(t) \tag{A.2}$$

According to Eq. (11) and Eq. (A1), the coefficient of $O(t)$ can be calculated as follows

$$A_0 = \sum_{j=1}^{n} K_j u_j(\Delta t) = K_1 u_1 + R_2 - R_1 + \cdots + R_n - R_{n-1} = R_n \tag{A.3}$$

where $u_j$ and $R_j$ denote $u_j(\Delta t)$ and $R_j(\Delta t)$, respectively. Then the coefficient of first order derivative of $O(t)$ can be derived as

$$A_1 = \sum_{j=2}^{n} K_j u_j(\Delta t) \sum_{r=1}^{j-1} K_r$$

$$= K_1 (R_2 - R_1) + (K_1 + K_2)(R_3 - R_2) \cdots + \sum_{r=1}^{n-1} K_r (R_n - R_{n-1})$$

$$= \sum_{r=1}^{n-1} K_r (R_n - R_r), \tag{A.4}$$

and the coefficient of second order derivative of $O(t)$ can be derived as





$$A_2 = \sum_{j=3}^{n} K_j u_j \left( \Delta t \right) \sum_{r_2 > r_1 = 1}^{j-1} K_{r_1} K_{r_2}$$

$$= K_1 K_2 \left( R_3 - R_2 \right) + \left( K_1 K_2 + K_1 K_3 + K_2 K_3 \right) \left( R_4 - R_3 \right) + \cdots + \sum_{r_2 > r_1 = 1}^{n-1} K_{r_1} K_{r_2} \left( R_n - R_{n-1} \right)$$

$$= \sum_{r_2 > r_1 = 1}^{n-1} K_{r_1} K_{r_2} \left( R_n - R_{r_2} \right) \tag{A.5}$$

Similarly, we can obtain:

$$A_p = \sum_{r_p > \cdots > r_1 = 1}^{n-1} \left( K_{r_1} \cdots K_{r_p} \right) \left( R_n - R_{r_p} \right) \tag{A.6}$$

Further, if the relation between $S_m$ and $R_m$, i.e., $S_m + R_m = 1$ is used, the coefficient $A_p$ can be calculated

by Eq. (13).

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
