# Peer review of "The Heterogeneous Discrete Generalized Nash Model for Flood Routing"

_Hydrology and Earth System Sciences, 2020_

## Referee Comment (RC1) · Anonymous Referee #1 · 6 May 2020

**General comments**

The authors present a new flood routing model, "HDGNM", which is a further development of the "DGNM" model, which was developed by the primary author. The authors expand on the DGNM model, a Nash-cascade model, by incorporating a heterogeneous S-curve. The motivation of the authors is to improve flow routing in rivers that exhibit changes in the slope and geometry along its reach. They apply their model for flow routing in a 105 km stretch of the Hanjiang River, and demonstrate that the HDGNM model provides smaller error statistics.

I want to preface this discussion by stating that I am not an expert on the mathematical development of Nash-cascade models, and I recommend to the editor to rely on a different referee to judge the novelty or necessity of this development within that branch

of study.

First, I want to complement the authors for presenting their study in such a concise format. Although the authors should expand on a few sections to ease understanding, the manuscript has a very respectable size. However, I would strongly recommend for the authors to let their manuscript be proofread by a native speaker.

I think my main comment on the manuscript is that, in its current state, I fail to see the benefits of the proposed approach. The quantified improvement over the DGNM model is well described in the case study (although it does seem a marginal improvement at best), but I'm not convinced that the entire approach is conceptually ill-conceived. This may in part be due to a lack of context or well defined objective in the introduction, but points to some deeper concern as well.

First a small note on the literature review. The authors discuss a wide arrange of literature (starting from Nash's original paper) from L23-L54 on Rainfall Runoff modelling, even though the manuscript focuses on flow routing. Only from L55 onward the authors turn toward the relevant literature. Perhaps a restructuring to better lead up to the main objective would be advisable. Regarding relevant literature, I feel the authors focus to much on the Nash-cascade types of models and developments thereof, at the expense of other state-of-the-art literature on distributed hydrological modelling (e.g. see Imhoff et al., 2020 and references therein. DOI:"10.1029/2019WR026807").

Second, if I had to distil an objective from this manuscript it would be (Paraphrasing from L16) "To adapt the DGNM for flow routing to better deal with river reaches with varying geometry". This objective overlooks other, perhaps better suited, methods to deal with flow routing in river reaches of varying geometry. Conceptually, I would expect models derived from the shallow water equations to provide strong competition indeed. A literature review discussing alternatives outside from Nash models, would help to persuade the reader that the proposed alternative is worthwhile.

Third, building on the previous section, I'm having trouble seeing the inherent conceptual benefit for the broader scientific community. Applying Nash models to flow routing in rivers like the Hanjiang is really stretching the conceptual interpretation of the model to (in my opinion) untenable limits. The authors state that introducing heterogeneity would theoretically improve the model, but this is not supported by a rigorous analysis of the physics of river flow that their modification tries to alleviate. An interesting addition could perhaps be found in discussing how, from a physical point of view, changes in slope and cross-sections are expected to influence travel times and distortion of the flood wave, highlighting the flaws in the DGNM and hypothesizing how the HDGNM addresses these flaws. In its current form, I lean toward seeing the HDGNM model as an (overly) complex data-based model, more akin to machine-learning models than to process-based models - which have their applications as well, but if seen as such, require proper introduction and review of relevant literature.

The case study itself is interesting and well defined, although some expansion on the case study (see specific comments) is required. The application of the HDGNM model is clear and results are well described, although somewhat marginal compared to DGNM. I would encourage the authors to publish the source code of their model and test data along-side the manuscript as well.

In summary, I think the manuscript needs extensive revision before publication in HESS would be advised - mainly to better place it in light of the state-of-the-art and highlight the academic advancement made. Although to be fair, I fear the inherent academic progress made by this manuscript, even if thoroughly revised as advised above, may remain too little to be considered for publication in HESS, and that a different journal may be better suited. I include some specific comments below, in the hope they will be useful to the authors.

**Specific comments**

L10: "The heterogeneous... the DGNM". This is very vague wording: I did not understand what the authors meant by 'conceptual interpretation of the DGNM' until much

later on. Consider rephrasing this.

L16: "The HDGNM ... change greatly". Be more specific (here, but certainly later in the manuscript) what is meant by 'greatly'

L53: "All of these ... Runoff modelling". Be more specific which improvements are relevant for the objective

L73: "The DGNM ... topograpy too". Check language

L83: what is a combination formula?

L106: "another way to deduce the HDGNM": what is the first? Why is another way required?

L111: "But for the basins with large topographical changes": Some form of conceptual sketch of what the authors mean by 'large topographical change' would be appreciated.

L125: it is unclear why this formula is introduced, nor how it follows from (4)

L133: what is a sub-river?

L150: "but it seems impossible..." I'm not sure I follow why it is supposed to seem impossible.

L211: please specify on what basis the river is subdivided into these reaches.

L211: please use scientific notation for the slopes (1.76*10**-4)

L212: It is indicated by whom?

L213: please make clear what sub-reaches 1 and two are (Huangjiagan-Guanghua and Guanghua-Taipingdian?)

L219: The selection criteria of floods should be better described. Are these all the flood waves that fulfil the stated criteria? What do the authors mean by (delta t = 3)?

L221: I don't understand what the authors mean by 'the simulation effect'

L221: The forecast capability of HDGNM cannot be tested by comparing to DGNM. Improvement over DGNM can be tested, but any forecasting prowess should be based on evidence (measurements)

L223: please specify how 'flood data' was obtained and what it consists of.

L226: please specify which parameters were optimised

L229: It would be very helpful if the authors could expand on the outcome of their optimisation exercise. Specifically, assuming that n=3 is an optimised value, is this an expected value? The authors state the the HDGNM is better suited to deal with topographical change, and this case study indeed shows four subreaches, of which the first one has a shallower slope than the final three. So, based on this information, would n=2 not be a more expected value? Or perhaps n=4, based on the number of subreaches the authors divide the river into.

L267: "The heterogeneous ... the DGNM". I think the way this sentence is phrased does not help the author's case. Would 'The HDGNM was derived by implementing a heteregeneous S curve into the DGNM model' not be more to the point?

L295: What would constitute a reasonable request?

**Technical corrections**

Figure 2: The size of the labels is a bit small and difficult to read.

Figure 2: Please indicate which of these flood are the calibration events and which are the validation events

---

## Referee Comment (RC2) · Nilay Dogulu (Referee) · 14 Jun 2020

General Comments

The manuscript "The Heterogeneous Discrete Generalized Nash Model for Flood Routing" by Yan et al. presents a methodological improvement to the Discrete Generalized Nash Model (DGNM) developed for river flow routing by the lead author (Yan et al, 2015, in JoH). The proposed modification in the model aims to address the issue of representing spatial heterogeneity in rivers (mainly due to river topography). The new model's performance, called HDGNM ("H" for heterogeneous), is tested on a case study from China, the middle Hanjiang River – its channel slope varies greatly. The results from HDGNM are compared to that of HIUH (Li et al., 2008) with respect to

"Relative error of peak discharge" and "Nash–Sutcliffe efficiency coefficient".

In general, the manuscript has some potential which is limited and hidden in its current version. The theoretical value is not weak, however, it is not too strong either as it lacks proper conceptual as well as structural presentation of the research framework. Please see below the major issues (that I could identify as a non-expert in Nash models, like the Referee #1) all of which need to be thoroughly addressed before any further decision on the manuscript's future in HESS (which I believe will highly depend on how well and how much of its hidden potential will be unraveled through the revisions by the authors who are indeed the experts in Nash models).

The Title reads very nicely at a first glance. However, looking at the relevant literature, one can see that its style is very similar to the previously published papers of the author (on Generalized Nash Model, GNM – listed below). I strongly believe that bringing forward explicitly the novelty of the paper in a more attractive way through the title will benefit the paper immensely, especially in the long term. Please see my suggestion below.

- Yan et al. (2015, JoH): The generalized Nash model for river flow routing

- Yan et al. (2019, JoHE): Discretization of the Generalized Nash Model for Flood Routing

- And now in HESS Discussions: The Heterogeneous Discrete Generalized Nash Model for Flood Routing

- New title (recommended): "Addressing Spatial Heterogeneity in the Discrete Generalized Nash Model for Flood Routing: the case of Hanjiang River, China" (second part is optional)

The Abstract: A more coherent and appealing layout supported with a broader and insightful perspective on the literature can be adopted. The current text does not adequately reflect the wider scope of the study in terms of how the literature's progress

has shaped efforts on (conceptual) hydrological modelling, particularly for operational purposes. The manuscript, in general, lacks the perspective required to highlight its contribution to hydrology research and practice. This is a fundamental limitation making the manuscript too narrow for the audience of HESS, and should be addressed in the Introduction and the Conclusions parts as well. The potential implications of the research findings for hydrological modelling should be discussed in the manuscript thoroughly.

Research Question & Objectives: There must be a solid paragraph where one reads the research questions addressed within the scope of this study. A good overview (in the text format) is given on how the literature progressed over the years after Nash, 1957. Yet, the authors fail to mention the gaps in the literature and explicitly explain how their research align within the greater picture. Dear authors, please make sure to state the questions & objectives formulated for this research in a concise and fluid manner. It is also important that why the discrete generalized Nash model (DGNM, by Yan et al. 2019), but not others, is chosen for addressing the issue of spatial heterogeneity in IUH.

Literature review (P2, P3, P4): A table summarizing each cited work, for example, with the columns: (1) author-journal-publication year, (2) the type of application (rainfall-runoff modelling, river flow routing etc.), (3) the approach of effort towards making IUH semi-/distributed (e.g. replacing equal reservoirs with unequal ones vs. dividing watersheds into a number of subwatersheds), (4) visual representation of the proposed reservoir system (i.e. to show how the system is conceptualized), (5) case study, (6) reported performance (if available). It would be great if you can insert in a column a simple schematic of how the reservoir system is configured in each study (see column 4). This will prove extremely valuable for highlighting how and why your contribution contributes the literature on development of Nash's Instantaneous Unit Hydrograph theory. Hence, a new figure should be inserted (in the Methodology part) to show the configuration this study is based on). A further couple of lines can be added to explain

the limitations and gaps in the literature.

Layout: The current layout is not clear and the content is mixed in several (apart) sections. The "Case Study" section is too long as it covers both the case study description and the results. As a remedy, adopt the following changes:

- Open a new section called "Heterogeneous Discrete Generalized Nash Model (HDGNM)" which will include the parts "Conceptual interpretation of the DGNM", "Heterogeneous S curve", and "Derivation of the heterogeneous DGNM" as subsections. Also, shorten the text where appropriate.

- Open a new section called "Methodology" which will include: Case study (P10 L195 - P12 L220); Model comparison (P12 L221-222 & P13 L230-242); Model calibration (P12 L222 - P13 L230 & P14 L245-247)

- Open a new section named "Results and discussion" which will include: P14 L247 - P15 L265; Table 1 & Figure 2

Results and discussion: In its current form, there is a very limited presentation and discussion of results. The manuscript can greatly benefit from a thoroughly written discussion which integrates the results obtained from different models (definitely include HIUH in the comparison, even better the model by Wan et al, 2016 too). Building upon such robust discussion the authors could more reasonably justify the significance of their findings.

Conclusion: It is very weak. It doesn't address the results obtained at all. Writing of this part deserves the most critical attention. The authors are strongly encouraged to address key limitations of their study with possible recommendations.

Specific Comments

P1 L11. "conceptual interpretation of the DGNM" is a rather vague description for highlighting the methodological novelty in this (specific, and perhaps narrow) research contribution.

P2 L24-25. Please elaborate on the concept of the linear reservoir cascade with a focus on its physical interpretation for a catchment.

P4 L39-79. Please cite any references on how DGNM's performance compares with other models, and justify openly why DGNM was decided to be improved by incorporating HIUH.

P4 L79-80. Add a paragraph describing how the paper is structured.

P7 L131-132. "leads to" doesn't sound right here. This one long sentence can be replaced with these two sentences: "The DGNM is developed on the basis of the Nash's IUH. However, unlike HIUH, DGNM fails to address spatial heterogeneity when applied to . . . ".

P7 L132-134. Does the sentence refer to the introduction of HIUH into DGNM? The previous sentence is about DGNM. Reading the whole paragraph, one can take to mean that the matter is about improving DGNM by incorporating HIUH in its theoretical framework. If so, please add "into DGNM" before "can reflect". (Well, reading the next section I understand that indeed this is exactly what is meant.)

P12 L218-220. Please put a new figure showing the discharge data, preferably a time series plot, where the selected flood events are indicated.

P12 L219. What do you mean by "low proportion of the lateral inflows (time interval $\Delta t$ = 3h)"?

P12 L220 & P14 L245-247. 8 flood events for calibration, 2 flood events for validation: What is the basis of your calibration and validation data selection? What are their statistical properties? *** Very important note on the terminology: If your aim is to test model performance during calibration process (also called training), such data set is called "cross-validation". This set basically imitates the test set (also called validation or verification data) and used to avoid the issue of overfitting. On the other hand, the validation data set is needed to validate the model's performance after it's built, i.e. to

imitate the model in operation. \*\*\* So, it is not clear from the given text if the results presented under the name "validation" are indeed for validation, or cross-validation. Please clarify.

P12 L221-222. Include this information in the abstract and introduction, too. Also, it would be interesting to include HIUH (Li et al., 2008) in model comparison. Could you please add HIUH model in your comparative analysis between HDGNM and DGNM? To bring variety in terms of types of models, you should also compare the results with a model (e.g. Wang et al., 2016) that adopts the second approach (i.e. dividing the watershed into sub-watersheds).

P12 L225-2230. The optimization procedure has not been explained adequately. Please describe the parameters optimized for these two models, and justify the logic behind the selected objective function (L226-227). Also, give examples of references where SCE-UA has been used for optimization of hydrological model parameters (and how its performance compares to other optimization methods.)

P13 L235-242. Model evaluation metrics: Please justify the reasons behind your selection of the error measures, if possible citing relevant papers in the literature. What are the weaknesses and strengths of these measures? What do their magnitude imply? Please add explanations.

P13-14 Table 1. Please convey the information graphically where the comparison can be visually made much more easily.

Minor Edits

- P1 L8. developed > recently developed

- P1 L17. The middle Hanjiang River > The middle Hanjiang River in China

- P1 L18. It is not appropriate to use "suggested" here. What comes next is the finding of your study. Better to simply use "found" or, "The results show that" . Also, "performs better" sounds rather vague – instead: "The HDGNM outperforms the DGNM in terms

of model efficiency and relative error . . ."

- P2 L27. It would be good to refer the readers to "Dooge, J.C.I., O'Kane, J.P., 2003. Deterministic Methods in Systems Hydrology. A.A. Balkema Publishers, Swets and Zeitlinger B.V., Lisse, The Netherlands." for further details on IUH theory.

- P3 L46. to consider > to account for

- P4 L73-74. Revise the sentence (it is grammatically incorrect).

- P7 L130. to the basin with > to basins with

- P8 L145-1466. K > K (It should be written in italic, right? Please be consistent throughout the manuscript.)

- P12 Figure 1. Please enlarge the figure, it is too small.

- P15 Figure 2. The resolution is poor, please increase the quality of the figure.

- References cited reflect the literature on Generalized Nash Model well. Citations are appropriately made. Only check the publication year of the reference: "Kalinin, G. P., and Milyukov, P. I.: On the computation of unsteady flow in open channels, Leningrad, Russia, Meteor. Gidrol. Zh., 10, 10–18, 1957." – It is cited in the text as 1958.

Cited references in the comment

- Li, C., Guo, S., Zhang, W., and Zhang J.: Use of Nash's IUH and DEMs to 345 identify the parameters of an unequal-reservoir cascade IUH model, Hydrol. Process., 22(20), 4073-4082. 2008.

- Wan, H., Xia, J., Zhang, L., Zhang, W., and Xu, C. Y.: A generalized concentration curve (gcc) method for storm flow hydrograph prediction in a conceptual linear reservoir-channel cascade, Hydrol. Res., 47(5), 932-950, 2016.

- Yan, B., Guo, S., Liang, J. and Sun., H.: The generalized Nash model for river flow routing, J. Hydrol., 530, 79-86, 2015.

- Yan, B., Huo, L., Liang, J., Yang, W., and Zhang, J.: Discretization of the generalized Nash model for flood routing, J. Hydrol. Eng., 24 (9), 04019029, 2019.

---

## Author Comment (AC1) · 10 Aug 2020

**Responses to Anonymous Referee #1**

The authors present a new flood routing model, "HDGNM", which is a further development of the "DGNM" model, which was developed by the primary author. The authors expand on the DGNM model, a Nash-cascade model, by incorporating a heteregeneous S-curve. The motivation of the authors is to improve flow routing in rivers that exhibit changes in the slope and geometry along its reach. They apply their model for flow routing in a 105 km stretch of the Hanjiang River, and demonstrate that the HDGNM model provides smaller error statistics.

I want to preface this discussion by stating that I am not an expert on the mathematical development of Nash-cascade models, and I recommend to the editor to rely on a different referee to judge the novelty or necessity of this development within that branch of study.

Reply: We very much appreciate the careful reading of our manuscript and valuable suggestions of the reviewer. According to the reviewer's helpful suggestion, we have rewritten the introduction and added the purpose and main contribution.

The goal of this paper is to take into consideration spatial heterogeneity in DGNM, which is a river flood routing model that accounts for the separate contributions of old water and new water. The spatial heterogeneity of underlying surface is one main source of the nonlinearity of the hydrological processes and a key factor restricting the development of the hydrological model, but it is also an important breakthrough to improve the forecast accuracy. The topographic heterogeneity of rivers has great effects on river flood routing. Most hydrologic routing models have not considered such spatial heterogeneity, which restricts the application of this type models, especially in river reaches where the river slopes and cross-sections change greatly. It is beneficial to take into consideration such spatial heterogeneity. DGNM based on the linear cascade concept has the potential to address heterogeneity. Fortunately, the explicit expression of DGNM addressing the spatial heterogeneity is obtained by strictly mathematical derivation and conceptual interpretation of the routing system.

First, I want to complement the authors for presenting their study in such a concise format. Although the authors should expand on a few sections to ease understanding, the manuscript has a very respectable size. However, I would strongly recommend for the authors to let their manuscript be proofread by a native speaker.

Reply: We have adjusted the layout of the manuscript, as well, the manuscript was carefully reread to check for language issues. We have replaced the initial mistakes and edited the sentences carefully.

According to the two reviewers' suggestions, the layout is adjusted as follows:
Section 1. Introduction

Section 2. Heterogeneous Discrete Generalized Nash Model
This new section includes the parts "Conceptual interpretation of the DGNM", "Heterogeneous S curve", and "Derivation of the heterogeneous DGNM" as subsections.

Section 3. Application
This section includes: Case study, and Model calibration.

Section 4. Results and discussion

Section 5. Conclusions

I think my main comment on the manuscript is that, in its current state, I fail to see the benefits of the proposed approach. The quantified improvement over the DGNM model is well described in the case study (although it does seem a marginal improvement at best), but I'm not convinced that the entire approach is conceptually ill-conceived. This may in part be due to a lack of context or well defined objective in the introduction, but points to some deeper concern as well.

Reply: Most hydrologic routing models are lumped, and hence fail to describe the spatial heterogeneity of the river reach. Developed from the linear cascade concept, the DGNM has the potential to take into account such spatial heterogeneity. The heterogeneous linear cascade concept is introduced in the DGNM. Based on the water balance equation for each reservoir and the hydraulic relation between cascade reservoirs as well as the conceptual interpretation of the DGNM, the heterogeneous DGNM (HDGNM) is fortunately deduced in an explicit expression of heterogeneous S curve and inputs and outputs. The HDGNM is strictly deduced in mathematics and has a rigorous conceptual interpretation of the routing process. That is the main contribution of the manuscript. To make it much clear, we have rewritten this part.

First a small note on the literature review. The authors discuss a wide arrange of literature (starting from Nash's original paper) from L23-L54 on Rainfall Runoff modelling, even though the manuscript focuses on flow routing. Only from L55 onward the authors turn toward the relevant literature. Perhaps a restructuring to better lead up to the main objective would be advisable. Regarding relevant literature, I feel the authors focus to much on the Nash-cascade types of models and developments thereof, at the expense of other state-of-the-art literature on distributed hydrological modelling (e.g. see Imhoff et al., 2020 and references therein. DOI:"10.1029/2019WR026807").

Reply: Thanks for the reviewer's suggestions. We have rewritten the introduction and added the developments of some distributed hydrological modeling in deal with the spatial heterogeneity.
In this part, with respect to the spatial heterogeneity exhibited in hydrological processes, we mainly focus on the way to deal with the spatial heterogeneity in conceptual models.

Discretization of watershed is common practice to transform the lumped conceptual models to semi-distributed and distributed models. Physically based discretization and conceptual discretization are summarized as two discretization approaches, and this manuscript is based on the second approach.

The added part of introduction is as follows:

"*The hydrological processes usually exhibit substantial spatial heterogeneity. That might be due to the spatial heterogeneity of rainfall and underlying surface. Spatial heterogeneity of a river basin increases the predicting uncertainty of streamflow using hydrological models (Adhikary et al., 2019), it is the key factor restricting but also promoting the development of hydrologic models. In the early hydrological modelling, a watershed is considered as a single unit for computations where the watershed parameters and variables are averaged over this unit (Dwarakish and Ganasri, 2015). These lumped models fail to describe the spatial heterogeneity of inputs, parameters, and processes. To better describe such spatial heterogeneity, the hydrological models have been developed from lumped to semi-distributed and distributed. It is common practice to transform the lumped conceptual models to semi-distributed and distributed models by discretizing the watershed into sub-watersheds (Arnold et al., 1990), grid cells (Liang et al., 1994; Vertessy and Elsenbeer, 1999; Yao et al., 2009), representative elementary watershed (Reggiani et al., 1998; Reggiani and Rientjes, 2005), hydrological response units (Arnold et al. 1998a, b), and so on. Such transition enables the model to take into account the spatial heterogeneity. However, not like the physically based models, e. g. SHE (Abbott et al., 1986a, b), whose most parameters can be measured in the field (observable), without some kind of calibration, much parameters of the conceptual models need to be calibrated and hence may cause the overparameterization and equifinality problems (Imhoff et al, 2020). To overcome such problems, Samaniego et al. (2010) introduced a multiscale parameter regionalization (MPR) to obtain seamless parameters across scales, where upscaling operators are used to aggregate catchment characteristics at a very detailed resolution to the modeling resolution and then a transfer function is applied to these catchment characteristics to calculate hydrological model parameters. It is an effective technique to integrate the spatial heterogeneity of physiographic characteristics. As a result, the number of parameters to be calibrated can be greatly reduced. To further lower the number of calibrated parameters in the MPR methodology, Imhoff et al. (2020) investigate the applicability of (pedo) transfer functions (PTFs) in combination with suitable upscaling operators for deriving seamless hydrologic model parameters. These PTFs are derived from laboratory experiments with point-scale samples in a bottom-up approach (see Van Looy et al., 2017), and are not constrained to the model but to actual field measurements, thus requiring no further model calibration. Contrasted to such bottom-up approach, Tran et al. (2018) proposed an alternative top-down approach by disaggregating parameters to higher resolutions using catchment properties and their spatial heterogeneity. This disaggregation approach is promising and can serve as an alternative to regionalization techniques. Either in a bottom-up or*

*top-down approach, the spatial heterogeneity of catchment characteristics is represented by relationships between the spatial distribution of parameter values and them."*

Second, if I had to distil an objective from this manuscript it would be (Paraphrasing from L16) "To adapt the DGNM for flow routing to better deal with river reaches with varying geometry". This objective overlooks other, perhaps better suited, methods to deal with flow routing in river reaches of varying geometry. Conceptually, I would expect models derived from the shallow water equations to provide strong competition indeed. A literature review discussing alternatives outside from Nash models, would help to persuade the reader that the proposed alternative is worthwhile.

Reply: Similar to the distributed hydrologic models, physically based hydraulic flood routing approach can significantly account for the spatial heterogeneity and should be an appropriate method to deal with flow routing in river reaches of varying geometry. However, the detailed channel geometry information required in these hydraulic models are difficult to obtain in some rivers. As a result, the simplified hydrologic routing methods such as Muskingum method and IUH method are usually used as an alternative in these rivers. In fact, the comparison to other methods, including the widely used Muskingum method and dynamic wave model (DWM), has been made in Yan et al. (2019). The results show that the DGNM can provide comparable (to DWM) or even better results (to Muskingum). The HDGNM is a modification of DGNM, and we don't make such repeated comparison any more.

Third, building on the previous section, I'm having trouble seeing the inherent conceptual benefit for the broader scientific community. Applying Nash models to flow routing in rivers like the Hanjiang is really stretching the conceptual interpretation of the model to (in my opinion) untenable limits. The authors state that introducing heterogeneity would theoretically improve the model, but this is not supported by a rigorous analysis of the physics of river flow that their modification tries to alleviate. An interesting addition could perhaps be found in discussing how, from a physical point of view, changes in slope and cross-sections are expected to influence travel times and distortion of the flood wave, highlighting the flaws in the DGNM and hypothesizing how the HDGNM addresses these flaws. In its current form, I lean toward seeing the HDGNM model as an (overly) complex data-based model, more akin to machine-learning models than to process-based models - which have their applications as well, but if seen as such, require proper introduction and review of relevant literature.

Reply: In the introduction part, we have reviewed the modifications of some linear cascade based conceptual models in addressing the spatial heterogeneity. No matter the watershed is conceptually discretized into a cascade or parallel of unequal reservoirs or channels, the spatial heterogeneity of watershed or rainfall can be partially reflected, and hence improving the forecast accuracy in different degrees. The HDGNM is also a linear cascade based model addressing the spatial heterogeneity in a similar way, and

an improvement should be expected similarly. From a physical point of view, the storage coefficient K is a reservoir detention characteristic and has a physical meaning of travel time. Physically, the changes in slope and cross-sections are expected to influence travel times and distortion of the flood wave. This influence can then be reflected by the storage coefficient K. In the DGNM, all linear reservoirs have a same K, such abrupt changes of the slope and cross-sections can not be addressed. In contrast, with different K, the HDGNM is more adaptive to these changes. We have added this analysis in the revised discussion part.

As we have emphasized that the HDGNM is a conceptual model, whose parameter has a specific physical meaning. Not like the machine-learning models, it is potential to estimate these parameters by the relationships with other physical characteristics, such as the bed slope and length. We will study this possibility in the future.

The case study itself is interesting and well defined, although some expansion on the case study (see specific comments) is required. The application of the HDGNM model is clear and results are well described, although somewhat marginal compared to DGNM. I would encourage the authors to publish the source code of their model and test data along-side the manuscript as well.

Reply: Thanks for the reviewer's positive comment for the application part. We would open the source code and data in the future.

In summary, I think the manuscript needs extensive revision before publication in HESS would be advised - mainly to better place it in light of the state-of-the-art and highlight the academic advancement made. Although to be fair, I fear the inherent academic progress made by this manuscript, even if thoroughly revised as advised above, may remain too little to be considered for publication in HESS, and that a different journal may be better suited. I include some specific comments below, in the hope they will be useful to the authors.

Reply: We have tried our best to revise the manuscript. The layout has been re-organized, especially the derivation of the model has been rewritten to make the contribution and novelty much clearer. The introduction has also been rewritten focusing on the spatial heterogeneity in conceptual hydrological models to reflect a wider scope of the study, which will be benefit for the broader scientific community. River flow routing is one of the key components of the hydrological modeling, and hydrologic routing is still an important way. Most of the recent hydrologic routing models are still lumped that cannot reflect the spatial heterogeneity of the river reaches. The DGNM based on the linear cascade concept has the potential to account for such spatial heterogeneity by replacing the equal linear reservoirs into unequal ones. However, it seems impossible to obtain the solution of high order differential equation of the Nash cascade model with unequal storage parameters directly. In this manuscript, the strict mathematical derivation is combined with the deeper conceptual interpretation

of IUH, S-curve and linear cascade as well as the DGNM to obtain the HDGNM. This integration is a perfect solution of the above difficult problem. The application results also proves the improvement of the proposed approach. We hope that the revised manuscript can be acceptable.

**Specific comments**

L10: "The heterogeneous... the DGNM". This is very vague wording: I did not understand what the authors meant by 'conceptual interpretation of the DGNM' until much later on. Consider rephrasing this.

Reply: In the revised abstract, this vague description has been replaced by "The discrete generalized Nash model (DGNM) based on the Nash cascade model has the potential to address spatial heterogeneity by replacing the equal linear reservoirs into unequal ones."

L16: "The HDGNM ... change greatly". Be more specific (here, but certainly later in the manuscript) what is meant by 'greatly'

Reply: The main purpose of this study is addressing the spatial heterogeneity in the DGNM. In the river flow routing system, the spatial heterogeneity is reflected in the greatly changed river slopes and cross-sections. Greatly changed is relative to the homogeneity, no specific scopes.

L53: "All of these ... Runoff modelling". Be more specific which improvements are relevant for the objective.

Reply: In the revised introduction, we have summarized a table (Table 1) to illustrate the contributions of each literature. The improvements of each model has been added too.

Table 1 Summary of the IUH based models in considering the spatial heterogeneity

| Authors | Models | Formulas | Visual representation |
|---|---|---|---|
| Dooge (1959) | Doogle IUH model | $q(t)=\dfrac{S}{T}\displaystyle\int_0^{t'\le T}\left(\dfrac{\delta(t-\tau)}{\prod\limits_{i=1}^{i(\tau)}(1+K_iD)}\right)\omega(\dfrac{\tau}{T})\,d\tau$ |  |
| Singh (1964) | Nonlinear IUH model | $q(t)=\dfrac{1}{(K_2-K_1)}\displaystyle\int_0^{t'\le T}\left[\exp(\dfrac{-(t-\tau)}{K_2})-\exp(\dfrac{-(t-\tau)}{K_1})\right]\omega(\tau)\,d\tau$ |  |
| Diskin et al. (1978) | Urban Parallel Cascade (UPC) IUH Model | $q(t)=\dfrac{\alpha_A}{K_A\Gamma(n_A)}\left(\dfrac{t}{K_A}\right)^{n_A-1}\exp\left(\dfrac{-t}{K_A}\right)+\dfrac{\alpha_B}{K_B(n_B-1)}\left(\dfrac{t}{K_B}\right)^{n_B-1}\exp\left(\dfrac{-t}{K_B}\right)$ |  |
| Bhunya et al. (2005) | Hybrid Model (HM) | $Q_2(t)=\dfrac{1}{(K_1-K_2)^2}\left[\left(t\exp\left(\dfrac{t}{K_1}\right)+t\exp\left(\dfrac{t}{K_2}\right)\right)-\dfrac{2K_1K_2}{(K_1-K_2)}\left(\exp\left(\dfrac{t}{K_1}\right)-\exp\left(\dfrac{t}{K_2}\right)\right)\right]$ |  |
| Singh et al. (2007) | Extended hybrid model (EHM) | $Q_2(t)=\dfrac{1}{(K_1-K_2)^2}\left|\exp\left(\dfrac{t-2T}{K_1}\right)\left[t-2\left(T+\dfrac{K_1K_2}{(K_1-K_2)}\right)\right]+\exp\left(\dfrac{(t-2T)}{K_2}\right)\left[t-2\left(T-\dfrac{K_1K_2}{(K_1-K_2)}\right)\right]\right|$ |  |

| | | | |
|---|---|---|---|
| Bhunya et al. (2008) | Two-reservoir variable storage coefficient (2VSC) model | $$Q_2(t) = ER_1\left\{\frac{1}{1-r}\left(\exp\left[-\frac{t-\Delta t}{K}\right]-\exp\left(-\frac{t}{K}\right)\right)+\frac{r}{1-r}\left[\exp\left(-\frac{t}{K_r}\right)-\exp\left(-\frac{t-\Delta t}{K_r}\right)\right]\right\}$$ |  |
| Li et al. (2008) | Heterogeneous IUH (HIUH) | $$q(t) = \sum_{j=1}^{n}\frac{K_j^{n-2}}{\prod_{i=1,i\neq j}^{n}(K_j-K_i)}e^{-\frac{t}{K_j}}$$ |  |
| Wang and Chen (1996) | Spatially distributed IUH | $$Q_k(t) = \sum_{L=0}^{m}\sum_{j=0}^{n}A(L,n-j+1)u(t-L\Delta t)\left[1-\sum_{i=0}^{j-1}(i!)^{-1}\left(\frac{t-L\Delta t}{k}\right)^i\exp\left(-\frac{t-L\Delta t}{k}\right)\right]$$ |  |
| Wan et al. (2016) | Generalized concentration curve (GCC) | $$Q_k(t) = \sum_{i=1}^{n}\sum_{t=1}^{tu_i}\left[I_i(k-t+1)\omega_i\sum_{j=1}^{i}\frac{K_j^{i-2}e^{-t-(i-1)\tau/K_j}}{\prod_{\substack{m=1\\m\neq j}}^{i}(K_j-K_m)}\right]$$ |  |

L73: "The DGNM ... topograpy too". Check language

Reply: We have rewritten the introduction part.

L83: what is a combination formula?

Reply: Here we have not explained accurately enough, it represents the calculation formula of the combinatorial number.

L106: "another way to deduce the HDGNM": what is the first? Why is another way required?

Reply: This expression is not accurate. It will be corrected in the revision text.

L111: "But for the basins with large topographical changes": Some form of conceptual sketch of what the authors mean by 'large topographical change' would be appreciated.

Reply: Large topographical change is relative to the homogeneity, no specific scopes. The schematic of the heterogeneous river flow routing system is illustrated in Fig. 1.

[Figure]

Fig. 1 Schematic of the heterogeneous river flow routing system

L125: it is unclear why this formula is introduced, nor how it follows from (4).

Reply: This formula is used in Appendix. We have removed it to the Appendix.

L133: what is a sub-river?

Reply: Sorry, the expression is not adequate. It should be "sub-reach".

L150: "but it seems impossible..." I'm not sure I follow why it is supposed to seem impossible.

Reply: Sorry, the expression is not adequate. Not like the outflow $O^{new}$, K can be replaced by $K_i$ directly. In the expression of $O^{old}$, K is not explicit in the equation, but implicit in the calculation of coefficients of $S_{n-j}$. Hence, $O^{old}$ cannot be obtained by directly replacing K to $K_i$.

L211: please specify on what basis the river is subdivided into these reaches.

Reply: In the cited reference, the subdivision is based on the location of the cities.

L211: please use scientific notation for the slopes (1.76*10**-4)

Reply: Will be corrected in the revised text.

L212: It is indicated by whom?

Reply: It is indicated by the cited reference.

L213: please make clear what sub-reaches 1 and two are (Huangjiagan-Guanghua and Guanghua-Taipingdian?)

Reply: Yes, sub-reaches 1 and 2 are the reaches Huangjiagan-Guanghua and Guanghua-Taipingdian, respectively. We will clarify this in the revision.

L219: The selection criteria of floods should be better described. Are these all the flood waves that fulfil the stated criteria? What do the authors mean by (delta t = 3)?

Reply: The expression is not adequate. It should be "the performance".

L221: I don't understand what the authors mean by 'the simulation effect'

Reply: The expression is not adequate. It should be "the performance".

L221: The forecast capability of HDGNM cannot be tested by comparing to DGNM. Improvement over DGNM can be tested, but any forecasting prowess should be based on evidence (measurements)

Reply: Yes, the expression is not adequate. Will be corrected in the revised text.

L223: please specify how 'flood data' was obtained and what it consists of.

Reply: Will be clarified in the revised text.

L226: please specify which parameters were optimized.

Reply: Will be clarified in the revised text.

L229: It would be very helpful if the authors could expand on the outcome of their optimisation exercise. Specifically, assuming that n=3 is an optimised value, is this an expected value? The authors state the the HDGNM is better suited to deal with topographical change, and this case study indeed shows four subreaches, of which the first one has a shallower slope than the final three. So, based on this information, would n=2 not be a more expected value? Or perhaps n=4, based on the number of subreaches the authors divide the river into.

Reply: Will be added in the revised text.

L267: "The heterogeneous ... the DGNM". I think the way this sentence is phrased does not help the author's case. Would 'The HDGNM was derived by implementing a heteregeneous S curve into the DGNM model' not be more to the point?

Reply: A good suggestion! Will be corrected in the revised text.

L295: What would constitute a reasonable request?

Reply: A reasonable request is usually about the purpose of the usage. It is not supported if the data was used as a commercial purpose.

**Technical corrections**
Figure 2: The size of the labels is a bit small and difficult to read.

Reply: Thank you for your suggestion. In the revised version we will take care to improve the readability of all tables and figures.

Figure 2: Please indicate which of these flood are the calibration events and which are the validation events

Reply: Will be corrected in the revised version.

---

## Author Comment (AC2) · 10 Aug 2020

**Responses to Referee #2 (Nilay Dogulu)**

In general, the manuscript has some potential which is limited and hidden in its current version. The theoretical value is not weak, however, it is not too strong either as it lacks proper conceptual as well as structural presentation of the research framework. Please see below the major issues (that I could identify as a non-expert in Nash models, like the Referee #1) all of which need to be thoroughly addressed before any further decision on the manuscript's future in HESS (which I believe will highly depend on how well and how much of its hidden potential will be unraveled through the revisions by the authors who are indeed the experts in Nash models).

Reply: We thank Referee #2 for his/her suggestions and comments, which helped improving the manuscript. To make the contribution and novelty of this manuscript much clearer, we accordingly restructured and re-formulated the relevant sections in the revision.

The goal of this paper is to take into consideration spatial heterogeneity in DGNM, which is a river flood routing model that accounts for the separate contributions of old water and new water. The spatial heterogeneity of underlying surface is one main source of the nonlinearity of the hydrological processes and a key factor restricting the development of the hydrological model, but it is also an important breakthrough to improve the forecast accuracy. The topographic heterogeneity of rivers has great effects on river flood routing. Most hydrologic routing models have not considered such spatial heterogeneity, which restricts the application of this type models, especially in river reaches where the river slopes and cross-sections change greatly. It is beneficial to take into consideration such spatial heterogeneity. DGNM based on the linear cascade concept has the potential to address heterogeneity. Fortunately, the explicit expression of DGNM addressing the spatial heterogeneity is obtained by strictly mathematical derivation and conceptual interpretation of the routing system.

The Title reads very nicely at a first glance. However, looking at the relevant literature, one can see that its style is very similar to the previously published papers of the author (on Generalized Nash Model, GNM – listed below). I strongly believe that bringing forward explicitly the novelty of the paper in a more attractive way through the title will benefit the paper immensely, especially in the long term. Please see my suggestion below.
- Yan et al. (2015, JoH): The generalized Nash model for river flow routing
- Yan et al. (2019, JoHE): Discretization of the Generalized Nash Model for Flood Routing
- And now in HESS Discussions: The Heterogeneous Discrete Generalized Nash Model for Flood Routing

- New title (recommended): "Addressing Spatial Heterogeneity in the Discrete Generalized Nash Model for Flood Routing: the case of Hanjiang River, China" (second part is optional)

Reply: This is a good suggestion, we have modified the title of the manuscript as "Addressing Spatial Heterogeneity in the Discrete Generalized Nash Model for Flood Routing".

The Abstract: A more coherent and appealing layout supported with a broader and insightful perspective on the literature can be adopted. The current text does not adequately reflect the wider scope of the study in terms of how the literature's progress has shaped efforts on (conceptual) hydrological modelling, particularly for operational purposes. The manuscript, in general, lacks the perspective required to highlight its contribution to hydrology research and practice. This is a fundamental limitation making the manuscript too narrow for the audience of HESS, and should be addressed in the Introduction and the Conclusions parts as well. The potential implications of the research findings for hydrological modelling should be discussed in the manuscript thoroughly.

Reply: Thanks for the reviewer's good suggestions. We have rewritten the abstract part as follows:

*"The spatial heterogeneity of rainfall and underlying surface is one main source of the nonlinearity of the hydrological processes and a key factor restricting the development of the hydrological model, but it is also an important breakthrough to improve the forecast accuracy. Except for the explicit watershed discretization approach, taking into account the spatial heterogeneity in the conceptualization is an alternative. River flood routing is one of the key components of hydrologic modeling. The topographic heterogeneity of rivers has great effects on river flood routing. It is beneficial to take into consideration such spatial heterogeneity, especially for hydrologic routing models. The discrete generalized Nash model (DGNM) based on the Nash cascade model has the potential to address spatial heterogeneity by replacing the equal linear reservoirs into unequal ones. However, it seems impossible to obtain the solution of this complex high order differential equation directly. Alternatively, the strict mathematical derivation is combined with the deeper conceptual interpretation of the DGNM to obtain the heterogeneous DGNM (HDGNM). The HDGNM is explicitly expressed as a linear combination of the inflows and outflows, whose weight coefficients are calculated by the heterogeneous S curve. The HDGNM expands the application scope, and becomes more applicable, especially in river reaches where the river slopes and cross-sections change greatly. The middle Hanjiang River was selected as a case study to test the model performance. It is suggested that the HDGNM performs better than the DGNM, with higher model efficiency and smaller relative error in the simulated flood hydrographs."*

Research Question & Objectives: There must be a solid paragraph where one reads the research questions addressed within the scope of this study. A good overview (in the text format) is given on how the literature progressed over the years after Nash, 1957. Yet, the authors fail to mention the gaps in the literature and explicitly explain how their research align within the greater picture. Dear authors, please make sure to state the questions & objectives formulated for this research in a concise and fluid manner. It is also important that why the discrete generalized Nash model (DGNM, by Yan et al. 2019), but not others, is chosen for addressing the issue of spatial heterogeneity in IUH.

Reply: According to the reviewer's suggestion, we have made a separate paragraph at the end of the introduction part to address the research question and the objective. This paragraph is written as follows:

*"Hydrologic routing approach has made great contributions to the development of conceptual hydrological models, and is still a widely used approach in river flood routing. However, most hydrologic routing methods are lumped and fail to reflect the spatial heterogeneity of the river reach. The development of IUH suggests that it is feasible to take into account spatial heterogeneity in IUH by the conceptual discretization of the watershed. The DGNM based on IUH has the potential to address spatial heterogeneity by replacing the equal linear reservoirs into unequal ones. However, the differential equation of the river flow routing system will become a complex high order differential equation and cannot be solved directly. A new way with the integration of mathematical derivation and conceptual interpretation of the DGNM is proposed to obtain the heterogeneous DGNM (HDGNM)."*

Literature review (P2, P3, P4): A table summarizing each cited work, for example, with the columns: (1) author-journal-publication year, (2) the type of application (rainfallrunoff modelling, river flow routing etc.), (3) the approach of effort towards making IUH semi-/distributed (e.g. replacing equal reservoirs with unequal ones vs. dividing watersheds into a number of subwatersheds), (4) visual representation of the proposed reservoir system (i.e. to show how the system is conceptualized), (5) case study, (6) reported performance (if available). It would be great if you can insert in a column a simple schematic of how the reservoir system is configured in each study (see column 4). This will prove extremely valuable for highlighting how and why your contribution contributes the literature on development of Nash's Instantaneous Unit Hydrograph theory. Hence, a new figure should be inserted (in the Methodology part) to show the configuration this study is based on). A further couple of lines can be added to explain the limitations and gaps in the literature.

Reply: This is a good suggestion. We have made an overview of each literature, and summarized each contribution in Table 1.

Table 1 Summary of the IUH based models in considering the spatial heterogeneity

| Authors | Models | Formulas | Visual representation |
|---|---|---|---|
| Dooge (1959) | Doogle IUH model | $q(t) = \dfrac{S}{T} \displaystyle\int_0^{t' \leq T} \left( \dfrac{\delta(t-\tau)}{\prod\limits_{i=1}^{i(\tau)} (1 + K_i D)} \right) \omega\!\left(\dfrac{\tau}{T}\right) \mathrm{d}\tau$ |  |
| Singh (1964) | Nonlinear IUH model | $q(t) = \dfrac{1}{(K_2 - K_1)} \displaystyle\int_0^{t' \leq T} \left[ \exp\!\left(\dfrac{-(t-\tau)}{K_2}\right) - \exp\!\left(\dfrac{-(t-\tau)}{K_1}\right) \right] \omega(\tau)\,\mathrm{d}\tau$ |  |
| Diskin et al. (1978) | Urban Parallel Cascade (UPC) IUH Model | $q(t) = \dfrac{\alpha_A}{K_A \Gamma(n_A)} \left(\dfrac{t}{K_A}\right)^{n_A - 1} \exp\!\left(\dfrac{-t}{K_A}\right) + \dfrac{\alpha_B}{K_B(n_B - 1)} \left(\dfrac{t}{K_B}\right)^{n_B - 1} \exp\!\left(\dfrac{-t}{K_B}\right)$ |  |
| Bhunya et al. (2005) | Hybrid Model (HM) | $Q_2(t) = \dfrac{1}{(K_1 - K_2)^2} \left[ \left( t\exp\!\left(\dfrac{t}{K_1}\right) + t\exp\!\left(\dfrac{t}{K_2}\right) \right) - \dfrac{2K_1 K_2}{(K_1 - K_2)} \left( \exp\!\left(\dfrac{t}{K_1}\right) - \exp\!\left(\dfrac{t}{K_2}\right) \right) \right]$ |  |
| Singh et al. (2007) | Extended hybrid model (EHM) | $Q_2(t) = \dfrac{1}{(K_1 - K_2)^2} \left| \exp\!\left(\dfrac{t - 2T}{K_1}\right) \left[ t - 2\left( T + \dfrac{K_1 K_2}{(K_1 - K_2)} \right) \right] + \exp\!\left(\dfrac{(t - 2T)}{K_2}\right) \left[ t - 2\left( T - \dfrac{K_1 K_2}{(K_1 - K_2)} \right) \right] \right|$ |  |

| | | | |
|---|---|---|---|
| Bhunya et al. (2008) | Two-reservoir variable storage coefficient (2VSC) model | $Q_2(t) = ER_1\left\{\dfrac{1}{1-r}\left(\exp\left[-\dfrac{t-\Delta t}{K}\right]-\exp\left(-\dfrac{t}{K}\right)\right)+\dfrac{r}{1-r}\left[\exp\left(-\dfrac{t}{K_r}\right)-\exp\left(-\dfrac{t-\Delta t}{K_r}\right)\right]\right\}$ |  |
| Li et al. (2008) | Heterogeneous IUH (HIUH) | $q(t) = \displaystyle\sum_{j=1}^{n}\dfrac{K_j^{n-2}}{\displaystyle\prod_{i=1,i\neq j}^{n}\left(K_j-K_i\right)}e^{-\frac{t}{K_j}}$ |  |
| Wang and Chen (1996) | Spatially distributed IUH | $Q_k(t) = \displaystyle\sum_{L=0}^{m}\sum_{j=0}^{n}A(L,n-j+1)\,u(t-L\Delta t)\left[1-\sum_{i=0}^{j-1}(i!)^{-1}\left(\dfrac{t-L\Delta t}{k}\right)^i\exp\left(-\dfrac{t-L\Delta t}{k}\right)\right]$ |  |
| Wan et al. (2016) | Generalized concentration curve (GCC) | $Q_k(t) = \displaystyle\sum_{i=1}^{n}\sum_{t=1}^{tu_i}\left[I_i(k-t+1)\omega_i\sum_{j=1}^{i}\dfrac{K_j^{i-2}e^{-t-(i-1)\tau/K_j}}{\displaystyle\prod_{\substack{m=1\\m\neq j}}^{i}\left(K_j-K_m\right)}\right]$ |  |

In the introduction part, with respect to the spatial heterogeneity exhibited in hydrological processes, we mainly focus on the way to deal with the spatial heterogeneity in conceptual models. Discretization of watershed is common practice to transform the lumped conceptual models to semi-distributed and distributed models. Physically based discretization and conceptual discretization are summarized as two discretization approaches. The literatures reviewed in Table 1 are based on the second approach. We have revised this part as follows:

*"Except for the physically based discretization approach, discretization of watershed conceptually is an alternative to account for spatial heterogeneity. For example, in flow routing, watershed is usually conceptualized as a cascade of equal linear reservoirs. Under this assumption, the instantaneous unit hydrograph (IUH) is obtained in a form of gamma distribution with two parameters - n, the number of linear reservoirs, and K, reservoir storage coefficient (Nash, 1957), and becomes one of the most widely used flow routing method. However, as a lumped model, IUH cannot reflect the spatial heterogeneity of rainfall and landforms. Conceptual discretization of watershed has been made to make IUH be semi-distributed or distributed by replacing the equal reservoirs with unequal ones or the combination with other components. Table 1 provides a snapshot overview of such modifications of the IUH to take into consideration spatial heterogeneity. The watershed was conceptually discretized by Dooge (1959) as a combination of unequal linear reservoirs and linear channels. The outflow from the linear channel was represented by a time-area diagram which, together with outflow from the preceding sub-area, serves as the inflow to the linear reservoir. Singh (1964) discretized the watershed into two unequal linear reservoirs and one linear channel considering the overland and channel flow components separately. To solve the flow routing in urban areas, Diskin et al. (1978) proposed an urban parallel cascade IUH model by representing the basin system as the combination of two parallel branches having a series of equal linear reservoirs. One of its most striking features is that it estimates separately the contribution of the impervious area of the watershed and that of the pervious area, and is thus useful for deriving flood hydrographs both for existing conditions in a watershed and for proposed changes in the degree of urbanization (Singh et al., 2014). Bhunya et al. (2005) developed a hybrid model by splitting the single linear reservoir into two serially connected reservoirs of unequal storage coefficients (one hybrid unit), and obtained the analytical expression of the model for two hybrid units in series. The hybrid model with two serially connected units is found to work significantly better than Nash's IUH. Later, to consider the translation time, Singh et al. (2007) extended this hybrid model by inserting a linear channel between each hybrid unit, which makes the extended hybrid model work significantly better than the hybrid model for large catchments. Bhunya et al. (2008) formulated a rainfall-runoff model incorporating a variable storage coefficient instead in the two-reservoir Nash cascade model. This model performs significantly better than the Nash's IUH in a small catchment but worse in a large catchment. Li et al. (2008) derived the IUH with different K values for each reservoir using the Laplace transform and developed a general rule for the equation of the IUH of any order, in which the*

*storage coefficient K is related to the slope of the main river channel. To consider the nonuniformly distributed rainfall, Wang and Chen (1996) divided the watershed into a number of sub-watersheds in series or parallel based on the structure of a stream network, and obtained the outflow hydrograph of each sub-watershed based on the concept of linear cascade reservoirs. This linear, spatially distributed model can be capable of predicting runoff from non-uniformly distributed rainfall and geographical conditions over an entire watershed. Similarly, Wan et al. (2016) divided the watershed into subareas by isochrones, and established an independent linear reservoir-channel cascade model in each subarea. The generalized concentration curve was derived to overcome the problem of heterogeneous rainfall distribution in larger watersheds. No matter the watershed is conceptually discretized into a cascade or parallel of unequal reservoirs or channels, the spatial heterogeneity of watershed or rainfall can be partially reflected, and hence improving the forecast accuracy in different degrees."*

In this study, the river flow routing system is conceptualized as a cascade unequal linear reservoirs. The schematic of this system has been figured out in the Methodology part, as shown in Fig. 1. Structurally, it is similar to that of HIUH in Li et al. (2008). The main difference is that as a river flow routing approach, HDGNM not only considers the contribution of the new water from upstream but also considers the contribution of the old water that stored at current time in river reach, which is a considerable component for river flow routing.

[Figure]

Fig. 1 Schematic of the HDGNM

As illustrated in Fig. 1, there are two numbering systems, named as "up numbering system" and "down numbering system" respectively. In the up numbering system, reservoirs are numbered from upstream to downstream, denoted by superscript "u". In the down numbering system, reservoirs are numbered from downstream to upstream, denoted by superscript "d". The linear relationship between the outflow and storage of each reservoir or each sub-reach is given in Fig. 1.

If the inflow I=$\delta$, where $\delta$ is the Dirac delta function, then the outflow O is the instantaneous unit hydrograph $u_n(t)$, and $u_n(t) = u_n^u(t) = u_n^d(t)$. The outflow of each reservoir is $u_i^u(t)$.

If the inflow I=1 all the time, then the outflow O is the corresponding S curve $S_n(t)$, and $S_n(t) = S_n^u(t) = S_n^d(t)$. Then the term $I_t S_n^u$ represents the outflow generated by the

constant inflow $I_t$. The outflow of each reservoir is $S_i^u$, then $K_i^u S_i^u$ represents the water stored in each reservoir for a unit continuous inflow, and $\sum_{i=1}^{n} K_i^u S_i^u / \Delta t$ represents the ratio of water stored in the channel during the period $\Delta t$. Then, $1 - \sum_{i=1}^{n} K S_i / \Delta t$ represents the ratio of water discharges from the channel. So $(1 - \sum_{i=1}^{n} K S_i / \Delta t) \Delta I_{t+1}$ is the outflow generated by the inflow increment during the time interval $[t, t + \Delta t]$ after the channel routing. Hence, the outflow generated by the new water (upstream inflow) is composed by the response of constant inflow $I_t$ and its increment during the time interval $[t, t + \Delta t]$, i. e.

$$O_{t+1}^{new} = S_n^u I_t + \left(1 - \frac{1}{\Delta t} \sum_{i=1}^{n} K_i^u S_i^u\right) \Delta I_{t+1}$$

The outflow generated by the new water is deduced by the conceptual interpretation of the concepts of IUH and its S curve under the up numbering system. Based on the DGNM, the downstream outflow is physically generated by the old water stored in the river reach and new water from the upstream inflow. It seems more convenient to obtain the outflow generated by the old water under the down numbering system. The storage routing equation of the j-th reservoir can be obtained from the water balance equation:

$$K_j^d \frac{dO_j^d(t)}{dt} = O_{j+1}^d(t) - O_j^d(t)$$

Based on the above equation, the outflow of each reservoir at the current time can be obtained as follows:

$$O_1^d(t) = O(t)$$

$$O_2^d(t) = O(t) + K_1^d O'(t)$$

$$O_3^d(t) = O(t) + \left(K_1^d + K_2^d\right) O'(t) + K_1^d K_2^d O''(t)$$

$$\ldots\ldots\ldots\ldots$$

$$O_j^d(t) = O(t) + \sum_{p=1}^{j-1} \prod_{r_p > \cdots > r_1 = 1}^{j-1} \left(K_{r_1}^d \cdots K_{r_p}^d\right) O^{(p)}(t)$$

Based on the physical interpretation of the GNM (Yan et al., 2015), the recession flow of the current water storage in river channel is the superposition of the recession flow generated by the current water storage in each reservoir. According to the conception of linear reservoir, the current water storage of the j-th reservoir is

$$W_j^d(t) = K_j^d O_j^d(t)$$

In the routing process, the volume of the water stored in each reservoir can be treated as an instantaneous inflow into each reservoir. As illustrated in Fig. 1, when I=$\delta$, the outflow is IUH $u_n(t)$. Here, the instantaneous inflow $I_j = \delta W_j^d(t)$, then the outflow at the end of the period generated by that is $W_j^d(t) u_j^d(\Delta t)$. Based on the principle of

superposition, the outflow at the end of the period formed by the current water storage of all reservoirs is

$$O_{t+1}^{old} = \sum_{j=1}^{n} K_j^d O_j^d (t) u_j^d (\Delta t)$$

$$= \sum_{j=1}^{n} K_j^d u_j^d (\Delta t) \left[ O(t) + \sum_{p=1}^{j-1} \sum_{r_p > \cdots > r_1 = 1}^{j-1} \left( K_{r_1}^d \cdots K_{r_p}^d \right) O^{(p)} (t) \right]$$

The formula shows that the recession process can finally be expressed as a linear combination of 0~(n-1) derivatives of the current time $O(t)$, which is

$$O_{t+1}^{old} = \sum_{p=0}^{n-1} A_p O_t^{(p)}$$

where $A_p (p = 0, \cdots, n-1)$ is the coefficient of p-th order derivative of $O(t)$, then we have (detailed derivation is provided in Appendix)

$$A_p = \begin{cases} 1 - S_n, \ p = 0 \\ \displaystyle\sum_{r_p > \cdots > r_1 = 1}^{n-1} \left( K_{r_1}^d \cdots K_{r_p}^d \right) \left( S_{r_p}^d - S_n \right), \ p > 0 \end{cases}$$

There is also an interesting finding that the no matter the reservoirs are numbered in the up numbering system or down numbering system, the results remain same. Hence, $A_p$ can be directly calculated in the up numbering system. Finally the outflow generated by the new water and old water under the same are obtained as heterogeneous DGNM, i. e.

$$O_{t+1} = \sum_{j=0}^{n-1} \sum_{i=0}^{j} \frac{(-1)^i}{\Delta t^j} C_j^i A_j O_{t-i} + S_n I_t + \left( 1 - \frac{1}{\Delta t} \sum_{i=1}^{n} K_i S_i \right) \Delta I_{t+1}$$

Layout: The current layout is not clear and the content is mixed in several (apart) sections. The "Case Study" section is too long as it covers both the case study description and the results. As a remedy, adopt the following changes:
- Open a new section called "Heterogeneous Discrete Generalized Nash Model (HDGNM)" which will include the parts "Conceptual interpretation of the DGNM", "Heterogeneous S curve", and "Derivation of the heterogeneous DGNM" as subsections. Also, shorten the text where appropriate.
- Open a new section called "Methodology" which will include: Case study (P10 L195-P12 L220); Model comparison (P12 L221-222 & P13 L230-242); Model calibration (P12 L222 - P13 L230 & P14 L245-247)
- Open a new section named "Results and discussion" which will include: P14 L247 - P15 L265; Table 1 & Figure 2

Reply: We have modified the layout of the manuscript according to your suggestions. The new layout is as follows.

Section 1. Introduction

Section 2. Heterogeneous Discrete Generalized Nash Model
This new section includes the parts "Conceptual interpretation of the DGNM", "Heterogeneous S curve", and "Derivation of the heterogeneous DGNM" as subsections.

Section 3. Application
This section includes: Case study, and Model calibration.

Section 4. Results and discussion

Section 5. Conclusions

Results and discussion: In its current form, there is a very limited presentation and discussion of results. The manuscript can greatly benefit from a thoroughly written discussion which integrates the results obtained from different models (definitely include HIUH in the comparison, even better the model by Wan et al, 2016 too). Building upon such robust discussion the authors could more reasonably justify the significance of their findings.

Reply: The comparison with other river flow routing method, such as the widely used Muskingum and dynamic wave model (DEM), has been made in Yan et al. (2019). The results show that the DGNM can provide comparable (to DWM) or even better results (to Muskingum). The HDGNM is a modification of DGNM, and we don't make such repeated comparison any more. HIUH and Wan's model (2016) are overland flow routing methods, they cannot be directly used in river flow routing, and hence the comparison to them cannot be given. From a physical point of view, changes in slope and cross-sections are expected to influence travel times and distortion of the flood wave. According to the suggestions of Reviewer 1, we have added the discussion about how these changes can be reflected in the HDGNM.

Conclusion: It is very weak. It doesn't address the results obtained at all. Writing of this part deserves the most critical attention. The authors are strongly encouraged to address key limitations of their study with possible recommendations.

Reply: We have rewritten this part. The results have been added in the revision. The key limitations of this study we think is how to find the relationships between the parameters and the physical characteristics, such as slop and length of the sub-reach, which deserves further study in the future.

Specific Comments
P1 L11. "conceptual interpretation of the DGNM" is a rather vague description for

highlighting the methodological novelty in this (specific, and perhaps narrow) research contribution.

Reply: In the revised abstract, this vague description has been replaced by "The discrete generalized Nash model (DGNM) based on the Nash cascade model has the potential to address spatial heterogeneity by replacing the equal linear reservoirs into unequal ones."

P2 L24-25. Please elaborate on the concept of the linear reservoir cascade with a focus on its physical interpretation for a catchment.

Reply: The concept of a linear reservoir cascade was originally proposed by Nash (1957) to model the runoff routing based on the assumption that the operation performed by a catchment on effective rainfall is analogous to that performed by routing through a series of linear reservoirs.

P4 L39-79. Please cite any references on how DGNM's performance compares with other models, and justify openly why DGNM was decided to be improved by incorporating HIUH.

Reply: The comparison with other river flow routing method, such as the widely used Muskingum and dynamic wave model (DEM), has been made in Yan et al. (2019). The results show that the DGNM can provide comparable (to DWM) or even better results (to Muskingum). The DGNM based on the Nash cascade model has the potential to address spatial heterogeneity by replacing the equal linear reservoirs into unequal ones.

P4 L79-80. Add a paragraph describing how the paper is structured.

Reply: We will add it in the revised manuscript.

P7 L131-132. "leads to" doesn't sound right here. This one long sentence can be replaced with these two sentences: "The DGNM is developed on the basis of the Nash's IUH. However, unlike HIUH, DGNM fails to address spatial heterogeneity when applied to : : : ".

Reply: Agree. We have rewritten this sentence.

P7 L132-134. Does the sentence refer to the introduction of HIUH into DGNM? The previous sentence is about DGNM. Reading the whole paragraph, one can take to mean that the matter is about improving DGNM by incorporating HIUH in its theoretical framework. If so, please add "into DGNM" before "can reflect". (Well, reading the next section I understand that indeed this is exactly what is meant.)

Reply: Yes, it refers to the introduction of HIUH into DGNM. We have revised this sentence.

P12 L218-220. Please put a new figure showing the discharge data, preferably a time series plot, where the selected flood events are indicated.

Reply: In the case study, 10 flood events not the long time series of the discharge were used to test the model. So we did not give this figure.

P12 L219. What do you mean by "low proportion of the lateral inflows (time interval _t= 3h)"?

Reply: The hydrologic routing method is based on the water balance equation of the river reach. This study mainly focuses on the river flood routing without considering the lateral inflows. A large lateral inflow would break this balance and influence the accuracy, so low proportion of the lateral inflows is required to test the model.

P12 L220 & P14 L245-247. 8 flood events for calibration, 2 flood events for validation: What is the basis of your calibration and validation data selection? What are their statistical properties? *** Very important note on the terminology: If your aim is to test model performance during calibration process (also called training), such data set is called "cross-validation". This set basically imitates the test set (also called validation or verification data) and used to avoid the issue of overfitting. On the other hand, the validation data set is needed to validate the model's performance after it's built, i.e. to imitate the model in operation. *** So, it is not clear from the given text if the results presented under the name "validation" are indeed for validation, or cross-validation. Please clarify.

Reply: The classification of the data is based on experience. Usually the ratio for training and verification is about 8:2 or 7:3. The "validation" in this study is for validation not cross-validation. We will clarify this in the revision.

P12 L221-222. Include this information in the abstract and introduction, too. Also, it would be interesting to include HIUH (Li et al., 2008) in model comparison. Could you please add HIUH model in your comparative analysis between HDGNM and DGNM? To bring variety in terms of types of models, you should also compare the results with a model (e.g. Wang et al., 2016) that adopts the second approach (i.e. dividing the watershed into sub-watersheds).

Reply: HIUH and Wan's model (2016) are overland flow routing methods, they cannot be directly used in river flow routing, and hence the comparison to them cannot be given. In the development of HDGNM, only the concept of HIUH as well as the corresponding heterogeneous S curve is used to deduce the HDGNM, but it is not a

separate component of the proposed model. In fact, the heterogeneous S curve is only used to express the coefficients of inflows and outflows in the HDGNM.

P12 L225-2230. The optimization procedure has not been explained adequately. Please describe the parameters optimized for these two models, and justify the logic behind the selected objective function (L226-227). Also, give examples of references where SCE-UA has been used for optimization of hydrological model parameters (and how its performance compares to other optimization methods.)

Reply: SCE-UA algorithm is a successfully proven method in global optimization (Duan et al. 1994) and has been extensively used in the hydrological model calibration. This algorithm is used to optimize parameters of the HDGNM by minimizing the root mean squared error (RMSE). We will give a detailed procedure for this optimization in the revised manuscript.

P13 L235-242. Model evaluation metrics: Please justify the reasons behind your selection of the error measures, if possible citing relevant papers in the literature. What are the weaknesses and strengths of these measures? What do their magnitude imply? Please add explanations.

Reply: A more detailed explanation about the metrics will be added in the revision.

P13-14 Table 1. Please convey the information graphically where the comparison can be visually made much more easily.

Reply: A good suggestion. We will revise the figures to make the comparison clearer.

Minor Edits
- P1 L8. developed > recently developed

Reply: Will be corrected in the revised text.

- P1 L17. The middle Hanjiang River > The middle Hanjiang River in China

Reply: Will be corrected in the revised text.

- P1 L18. It is not appropriate to use "suggested" here. What comes next is the finding of your study. Better to simply use "found" or, "The results show that" . Also, "performs better" sounds rather vague – instead: "The HDGNM outperforms the DGNM in terms of model efficiency and relative error : : :"

Reply: Will be corrected in the revised text.

- P2 L27. It would be good to refer the readers to "Dooge, J.C.I., O'Kane, J.P., 2003.

Deterministic Methods in Systems Hydrology. A.A. Balkema Publishers, Swets and Zeitlinger B.V., Lisse, The Netherlands." for further details on IUH theory.

Reply: A good suggestion. We will add the reference in the revised text.

- P3 L46. to consider > to account for

Reply: Will be corrected in the revised text.

- P4 L73-74. Revise the sentence (it is grammatically incorrect).

Reply: Will be corrected in the revised text.

- P7 L130. to the basin with > to basins with

Reply: Will be corrected in the revised text.

- P8 L145-1466. K > $K$ (It should be written in italic, right? Please be consistent throughout the manuscript.)

Reply: Will be corrected in the revised text.

- P12 Figure 1. Please enlarge the figure, it is too small.

Reply: Will be corrected in the revised text.

- P15 Figure 2. The resolution is poor, please increase the quality of the figure.

Reply: Will be corrected in the revised text.

- References cited reflect the literature on Generalized Nash Model well. Citations are appropriately made. Only check the publication year of the reference: "Kalinin, G. P., and Milyukov, P. I.: On the computation of unsteady flow in open channels, Leningrad, Russia, Meteor. Gidrol. Zh., 10, 10–18, 1957." – It is cited in the text as 1958.

Reply: Will be corrected in the revised text.